# Short-Term Effects of *Cenchrus fungigraminus*/Potato or Broad Bean Interplanting on Rhizosphere Soil Fertility, Microbial Diversity, and Greenhouse Gas Sequestration in Southeast China

**DOI:** 10.3390/microorganisms12081665

**Published:** 2024-08-13

**Authors:** Jing Li, Yufang Lei, Yeyan Wen, Jieyi Zhu, Xiaoyue Di, Yi Zeng, Xiao Han, Zuhui Que, Hatungimana Mediatrice, Christopher Rensing, Zhanxi Lin, Dongmei Lin

**Affiliations:** 1National Engineering Research Center of Juncao Technology, College of Juncao and Ecology, Fujian Agriculture and Forestry University, Fuzhou 350002, China; fafulijing@fafu.edu.cn (J.L.); 5220330057@fafu.edu.cn (Y.L.); 5220544015@fafu.edu.cn (Y.W.); 5220543016@fafu.edu.cn (J.Z.); 52305044024@fafu.edu.cn (X.D.); 52305043002@fafu.edu.cn (Y.Z.); mediatunga@gmail.com (H.M.); crensing94@gmail.com (C.R.); lzxjuncao@163.com (Z.L.); 2Shunchang Agriculture Science Research Institute, Nanping 353200, China; hanxiao_hx123@163.com; 3Zhengfang Rural Revitalization and Development Center of Shunchang, Nanping 353216, China; zfsnfwzx@163.com; 4Institute of Environmental Microbiology, College of Resource and Environment, Fujian Agriculture and Forestry University, Fuzhou 350002, China

**Keywords:** intercropping, *Cenchrus fungigraminus*, rhizosphere soil, short-term, methane and nitrogen cycling

## Abstract

*Cenchrus fungigraminus* is a new species and is largely used as forage and mushroom substrate. However, it can usually not be planted on farmland on account of local agricultural land policy. Interplanting *Cenchrus fungigraminus* with other crops annually (short-term) is an innovative strategy to promote the sustainable development of the grass industry in southern China. To further investigate this, *C. fungigraminus* mono-planting (MC), *C. fungigraminus*–potato interplanting (CIP) and *C. fungigraminus*–broad bean interplanting (CIB) were performed. Compared to MC, soil microbial biomass carbon (SMBC), soil organic matter (SOM), ammoniacal nitrogen (AMN), pH and soil amino sugars had a positive effect on the rhizosphere soil of CIP and CIB, as well as enhancing soil nitrogenase, nitrite reductase, and peroxidase activities (*p* < 0.05). Moreover, CIP improved the root vitality (2.08 times) and crude protein (1.11 times). In addition, CIB enhanced the crude fiber of *C. fungigraminus* seedlings. These two interplanting models also improved the microbial composition and diversity (Actinobacteria, Firmicutes, and Bacteroidota, etc.) in the rhizosphere soil of *C. fungigraminus* seedlings. Among all the samples, 189 and 59 genes were involved in methane cycling and nitrogen cycling, respectively, which improved the presence of the serine cycle, ribulose monophosphate, assimilatory nitrate reduction, methane absorption, and glutamate synthesis and inhibited denitrification. Through correlation analysis and the Mantel test, the putative functional genes, encoding functions in both nitrogen and methane cycling, were shown to have a significant positive effect on pH, moisture, AMN, SOM, SMBC, and soil peroxidase activity, while not displaying a significant effect on soil nitrogenase activity and total amino sugar (*p* < 0.05). The short-term influence of the interplanting model was shown to improve land use efficiency and economic profitability per unit land area, and the models could provide sustainable agricultural production for rural revitalization.

## 1. Introduction

Interplanting (or intercropping) is the simultaneous cultivation method of two or more crops in the same field for one or multiple years [1]. Compared with mono-planting, interplanting has long been practiced in both Chinese and worldwide agricultural history for efficient light interception, high crop yield, the phytoextraction of heavy metals, the introduction of microbial diversity, the enhancement of soil microbial activity, augmenting soil enzymatic activities, and improving soil fertility and nitrogen fixation in soil [2,3,4,5,6]. Meanwhile, interplanting not only maximizes the spatial value of cultivated land, but also fully utilizes solar energy, comprehensively improves the ventilation and light transmission conditions of field crops, and fully leverages the advantages of edge row yield increases. It has been shown to fully utilize and extend the crop growth season to achieve more than one year of maturity and enhanced harvest, and increase income per unit land on limited land areas. Due to the relatively large spacing between interplanting plants and second crop rows, it was able to effectively solve the issue of competition for water and fertilizer [7]. Plants harbor numerous microbial communities in the rhizosphere ecosystem. The positive interactions of these microbes with the plants increased rhizosphere soil fertility, while some of the microbes competed with plants for nutrients and other resources [8]. It had previously been established that soil microbial diversity has a critical role in maintaining the sustainability of agricultural production systems [9]. The soil microbial community was shown to maintain soil fertility and support sustainable plant growth and productivity by degrading organic matter [10]. Interplanting alters the microbial community structure and activity and thereby influences carbon and nitrogen dynamics [11,12]. Microbial communities also play an important role in every biogeochemical cycle and were shown to participate in various processes (nitrogen cycling, methane absorption, organic matter decomposition, soil agglomeration and humus formation, etc.), obtaining important mineral nutrients such as nitrogen, phosphorus, and sulfur from the soil; they were also shown to be major contributors to plant growth [13,14,15,16]. According to Lian et al. [17], sugarcane and soybean interplanting increased microbial diversity in rhizosphere soil and has been widely used to stabilize yield and reduce nitrogen leaching. Meanwhile, biological nitrogen fixation is one of the most important ecological and agricultural benefits of plant–bacterial interactions. Therefore, the recruitment of nitrogen-fixing bacteria in symbiotic or nonsymbiotic relationships help crops and plants obtain nitrogen directly from the atmosphere to meet their nutritional needs [18]. More than 80% of methane is produced through microbial activity and plays an important role in regulating climate change [19]. Some methane is absorbed and utilized by microorganisms before entering the atmosphere, and differences in substrate levels, physicochemical properties, osmotic pressure, and pH in soil determine the methane absorption capacity and diversity of methane-producing bacteria [20].

Currently, *Cenchrus fungigraminus* (previously referred to as *Pennisetum giganteum*, Jujuncao or Giant Juncao) is grown in 31 provinces in China and promoted in 106 countries worldwide as a new species [21]. It is a perennial C4 grass with high yield and photosynthesis [22] and is widely used in environmental management, mushroom cultivation, and as fodder, yielding good economic benefits for poverty alleviation and rural revitalization as part of the One Belt, One Road strategy [23,24,25]. However, because *C. fungigraminus* is a kind of forage grass imported from South Africa, it cannot be normally planted on farmland in accordance with local agricultural land policy. Therefore, the interplanting of *C. fungigraminus* with other crops may effectively solve problems in actual production practice. Grass interplanting has been demonstrated to be an effective system for orchard management [26]. Previous studies have described an increase in soil organic matter content [27,28] and water infiltration and a decrease in soil erosion and evaporation in such systems [29,30]. Grass interplanting also influences soil bacterial structure, microbial community diversity, and bacterial function, such as carbon metabolism-related activity [26]. The grass interplanting system was shown to specifically increase the topsoil total nitrogen, available phosphorus, and available potassium contents [31] and maintain the soil nitrogen content [32]. It also promoted the activity of soil enzymes, such as β-glucosidase, β-xylosidase, and cellobiohydrolase, involved in plant polysaccharide (cellulose and hemicellulose) hydrolysis [33]. However, current data on the effect of *C. fungigraminus* interplanting with other crops on production, microbial diversity and the economy are not available.

In this study, *C. fungigraminus* was interplanted with potato (*Solanum tuberosum*) and broad bean (*Vicia faba*), respectively. The correlation between the microbial community in the rhizosphere, soil properties, and nutrient levels requires a comprehensive analysis of interplanting models. By comparing the composition and function of rhizosphere microorganisms in the seedling period and yield, we determined (1) how *C. fungigraminus* interplanting changed soil fertilization, enzyme activities and greenhouse gas sequestration in one year; (2) how interplanting helped improve *C. fungigraminus* yield and forage quality; (3) how *C. fungigraminus* interplanting with crops changed the composition and diversity of the microbial community in the *C. fungigraminus’s* seedling period; and (4) the economic benefits to farmland in a short-term period. We hope this study will improve the understanding of the short-term dynamics on the rhizosphere soil of *C. fungigraminus* interplanting, which will be helpful for formulating sustainable and environmentally friendly agricultural management strategies.

## 2. Material and Methods

### 2.1. Experimental Design and Sample Collection

On-site field experiments of *C. fungigraminus* monoplanting (MC), *C. fungigraminus*–potato interplanting (CIP) and *C. fungigraminus*–broad bean interplanting (CIB) were conducted in Juncao Science and Technology Backyard in Shunchang County, Fujian Province (N 26°43′9″, E 117°42′12″, height 240 m, Figure 1a). The average annual temperature in the region was 18.5 °C. The annual total radiation was 4080–4780 Kcal·cm^−1^, the annual average sunshine time was 1740.7 h, and the frost-free period was 305 days. The annual rainfall in the entire region was approximately 1098 mm. The original experimental soil (OS) was basic farmland soil and had the characteristics described in Figure 2. *C. fungigraminus* was developed and supplied by the National Engineering Research Center of Juncao Technology. The potato seeds (Wotu No. 5) and broad bean seeds (Pu No. 10) from the locality market were used in this study. *C. fungigraminus* was planted in April and the aboveground part was harvested in October 2022, and the other two crops were planted in December 2022. For the OS, one monoculture (MC) and two interplanting patterns (CIP and CIB) were established for each plot (18 m × 30 m), and we created 6 replicates using the parallel sampling method. Three replicates were taken at each sample, and mixed before sequencing and testing. The experimental plots were planted with *C. fungigraminus* alternated with potato and broad bean at a 0.25 m distance (Figure 1b,c). After treatment, we recorded the yield of *C. fungigraminus*, broad bean and potato in the treatment plots. Standard agricultural practices for planting of *C. fungigraminus* and crops were followed, including covering with plastic film, irrigation, fertilization, weeding, and pest control. In our study, all samples were collected after potatoes and broad beans were harvested on 25 April 2023. *C. fungigraminus* yield was measured on 20 July 2023. Six rhizosphere soil samples of healthy *C. fungigraminus* seedlings were obtained by gently shaking large soil blocks and collecting the soil tightly attached to the root surface at a distance of approximately 1–20 mm. Six *C. fungigraminus* seedling leaves, stems, and roots were collected by using scissors. All soil and plant samples were taken to the laboratory on dry ice. One part of the samples was stored at −20 °C for use within 48 h for physicochemical property testing, while the other part was stored at −80 °C for use in high-throughput sequencing in the future. The experimentation ran from 2022 to 2023. The local climate conditions during the experiment from May 2022 to July 2023 are shown in Appendix A.

### 2.2. Analysis of Physicochemical Properties of Soil and Plants

Soil pH was determined using distilled water (*v*/*v*, 1:2.5) and analyzed by a pH meter (Sartorius, Göttingen, Germany) [34]. The soil moisture content (WC) was calculated using an oven (Jiangnan, Ningbo, China) in the laboratory. The soil nitrate nitrogen (NIN) was measured using the dual wavelength colorimetric method, and ammonium nitrogen (AMN) was measured using the indophenol blue colorimetric method according to the China national guideline LY/T 1228-2015 [35]. The soil organic matter (SOM) was measured following the potassium dichromate volumetric method [36]. Soil microbial biomass carbon (SMBC) was analyzed using the chloroform fumigation–extraction method [37]. The aboveground parts of plants were crushed and collected as random samples. The plant height, dry root weight and root activity in each group were recorded. The root activity was determined by the triphenyl tetrazolium chloride (TTC) method [7]. The total sugar (TS), crude protein (CP) and crude fiber (CF) content of *C. fungigraminus* were determined by using kits from Shanghai UPLC–MS Ltd. (Shanghai, China).

### 2.3. Determination of Soil Secondary Metabolites and Enzyme Activity

The four kinds of amino sugars in rhizosphere soil, including galactosamine (GalN), mannosamine (ManN), glucosamine (GlcN), and muramic acid (Mur), were detected on a GC-7890B (Agilent, Santa Clara, CA, USA, P-5MS capillary column 30 m (length) × 0.25 mm (diameter) × 0.25 μm (thickness)) with an injector temperature of 235 °C, a split ratio of 10:1, N_2_ as the carrier gas, and a flow rate of 1.2 mL·min^−1^ [38,39,40,41].

The soil catalase (S-CAT) activity was determined following the method of Chance and Maehly [42]. H_2_O_2_ (1 mmol) was dripped into 1 g of soil, and the enzyme activity was measured at 240 nm. Soil peroxidase (S-POD) activity detection was performed according to the method of Stevenson [40]. Soil nitrogenase (S-NITS) was detected by an ELSIA Kit (Solarbio, Beijing, China). Nitrogenase forms an antibody–antigen–enzyme-labeled antibody complex. After washing, the substrate 3,3′,5,5′-Tetramethylbenzidine (TMB) was added. TMB turns blue under the catalysis of horseradish peroxidase (HRP). The absorbance (OD value) was measured by a microplate reader at a wavelength of 450 nm. Soil nitrite reductase (S-NiR) is able to reduce nitrite ions to NO and other substances under anaerobic conditions, resulting in a reduction in nitrite ion content. Nitrite ions were first diazotized with p-aminobenzenesulfonic acid to form diazo compounds and then coupled with α-amine to form a purple red diazo dye, which could be detected by measuring the absorbance at 540 nm [43].

### 2.4. DNA Extraction and Metagenome Shotgun Sequencing

The total microbial genomic DNA in *C. fungigraminus* rhizosphere soil in six samples of each treatment (MC, CIP, and CIB) were extracted, respectively, using the OMEGA Mag-Bind Soil DNA Kit (Omega, San Clemente, CA, USA) following the manufacturer’s instructions, and stored at −20 °C prior to further assessment. The quantity and quality of the extracted DNA were measured using a Qubit™ 4 Fluorometer (Invitrogen, Carlsbad, CA, USA) and agarose gel electrophoresis. The extracted microbial DNA was processed to construct metagenome shotgun sequencing libraries with insert sizes of 400 bp by using an Illumina TruSeq Nano DNA LT Library Preparation Kit. Each library was sequenced by the Illumina NovaSeq platform (Illumina, San Diego, CA, USA) with the PE150 strategy at Personal Biotechnology Co., Ltd. (Shanghai, China). Adaptor sequences were removed from the raw sequencing reads using Cutadapt (v1.2.1) [44], and the reads were trimmed using a sliding-window algorithm in fastp [45]. Once quality-filtered reads were obtained, the taxonomic classification of the metagenomic sequencing reads from each sample was performed using Kraken2 [46] against a RefSeq-derived database. Reads assigned to metazoans or Viridiplantae were removed for downstream analysis. Megahit (v1.1.2) was used to assemble each sequence using the meta-large presented parameters [47]. The generated contigs (longer than 300 bp) were then pooled together and clustered using MMseqs2 [48] with “easy-linclust” mode, setting the sequence identity threshold to 0.95 and covering residues of the shorter contigs at 90%. The lowest common ancestor taxonomy of the nonredundant contigs was obtained by aligning them against the National Center for Biotechnology Information nucleotide sequence (NCBI-nt) database by MMseqs2 with “taxonomy” mode, and contigs assigned to Viridiplantae or Metazoa were dropped in the following analysis. The species were annotated by MMseqs2 according to the Genome Taxonomy Database (GTDB, https://gtdb.ecogenomic.org/), NCBI (https://www.ncbi.nlm.nih.gov/genbank/), Methane Cycling Database (MCycDB) (https://github.com/qichao1984/MCycDB) and Nitrogen Cycling Database (NCycDB, https://github.com/qichao1984/NCyc) databases, etc. The raw data derived from the Illumina platform were deposited in the NCBI database (https://www.ncbi.nlm.nih.gov/) with the accession number PRJNA1004187 (17 August 2023).

### 2.5. Statistical and Bioinformatics Analysis

Analysis of variance (ANOVA) or Mantel test was performed with a mean of six replicate values of each treatment data value, and Duncan’s multiple range test was employed to determine significant differences at 95% (*p* < 0.05) by SPSS Statistics 26 (IBM, Armonk, NY, USA) and Microsoft Office Excel 2010 (Microsoft, Redmond, SEA, USA), indicated by different lowercase letters. Based on the taxonomic and functional profiles of nonredundant genes, linear discriminant effect size (LEfSe) analysis was performed to detect differential abundant taxa and functions across the three treatment groups using the default parameters [49]. Beta diversity analysis was performed to investigate the compositional and functional variations of microbial communities across samples using the Bray–Curtis distance metric [50], and the results were visualized via principal coordinate analysis (PCoA) and hierarchical clustering using the unweighted pair-group method with arithmetic means (UPGMA) [51]. Redundancy analysis (RDA) and net heatmap generation were performed by GenesCloud Tools, a free online platform for data analysis (https://www.genescloud.cn). R studio 4.3.1 (Rstuido, Cambridge, MA, USA) was used to construct partial least square path models (PLS-PM) for the *C. fungigraminus* yield, soil physical and chemical properties, amino sugar level, soil microbial abundance, and genes involved in methane cycling and nitrogen cycling. All figures were generated using GraphPad Prism 9 (GraphPad Software, Boston, MA, USA), Adobe Photoshop 2020, and Adobe Illustrate 2021 (Adobe, San Jose, CA, USA).

## 3. Results

### 3.1. Analysis of Growth, Yield, Nutrition, and Economic Benefits

The total growth periods of *C. fungigraminus*, potato and broad bean were nearly 300 days, 110 days and 126 days, respectively. With the CIP and CIB treatments, the height and leaf length of *C. fungigraminus* seedlings decreased, and the tiller number increased, though the differences were not significant (*p* > 0.05). Meanwhile, the dry root weight of *C. fungigraminus* seedlings in MC was 2.69 times and 2.61 times higher than that of CIP and CIB, respectively (*p* < 0.05). In April 2023, the TS content of MC was 1.24 times and 1.11 times greater than that of CIP and CIB (*p* < 0.05). The CIP significantly increased the CP and CF content (*p* < 0.05) of *C. fungigraminus* seedlings compared to MC. The CIP significantly improved the vitality of *C. fungigraminus* seedling roots, which was 4.09 times and 2.08 times higher than that under CIB and MC (*p* < 0.05, Table 1). Until 20 July 2023, the plant height, leaf length and tiller number in MC were 359.67 ± 8.94 cm, 151.00 ± 8.94 cm, and 14.17 ± 2.04 and 352.33 ± 23.95 cm, 144.57 ± 6.56 cm, and 14.17 ± 3.97 in CIP and 338.67 ± 16.54 cm, 135.83 ± 8.42 cm and 14.50 ± 2.66 in CIB, respectively. The total values for MC, CIP and CIB were Chinese Yuan (CNY) 20,355, 23,412, and 20,600 per hectare, respectively (Table 1).

### 3.2. Physicochemical Properties and Amino Sugar Content of the Rhizosphere Soil of C. fungigraminus Seedlings

Compared with the soil before planting (OS), MC, CIP and CIB significantly decreased the soil pH (*p* < 0.05); MC was able to significantly increase the soil moisture content (*p* < 0.05, Figure 2a). The levels of NIN, SMBC, and SOM under the three planting modes were significantly improved compared to those under OS (*p* < 0.05). The levels of AMN, SMBC and SOM in CIP and CIB were significantly increased compared to those in MC (*p* < 0.05, Figure 2b). All the planting modes significantly improved the S-NiR (*p* < 0.05) to reduce the toxicity of nitrite in rhizosphere soil to *C. fungigraminus*. The S-POD activity was significantly improved by *C. fungigraminus* interplanting with broad bean compared to OS (*p* < 0.05, Figure 2c). The monoculture of *C. fungigraminus* significantly (*p* < 0.05) improved the activity of CAT in the soil at the seedling stage and exhibited a strong ability to degrade H_2_O_2_ in the soil, thus reducing its toxic effect on plants. However, in the interplanting mode, S-CAT activity was significantly reduced. According to Appendix A, the total amounts of the four amino sugars under OS, MC, CIP and CIB were 2.35 ± 0.11 mg·g^−1^, 3.85 ± 0.09 mg·g^−1^, 4.37 ± 1.59 mg·g^−1^, and 4.95 ± 0.74 mg·g^−1^, respectively; of these, the Mur content was the highest among the four components (Appendix A).

### 3.3. Microbial Community Structure and Diversity of C. fungigraminus Seedlings’ Rhizosphere Soil

We obtained a total of 115.38 GB of raw data, with an average base sequencing accuracy of 93.48%. A total of 112.51 GB of clean data were collected after filtering and quality control. The effective sequence ratios annotated to the genus level in the CIB, CIP and MC samples were 5.09%, 4.49%, and 3.99%, respectively, by the Kaiju annotation method [52] (Appendix A). The reads with a k-mer depth less than 2 were deleted, and the wrong base was corrected; then, the assembly was spliced by Megahit. The average N50 values of CIB, CIP and MC were 671, 540, and 641, respectively, and the open reading frame sizes were 89,192, 93,833 and 123,406, respectively. Using minimap2 to compare the effective read length sequence with the nonredundant overlapping group sequence obtained by splicing, the average alignment rates of CIB, CIP, and MC were 5.79%, 3.00%, and 4.86%, respectively. The species annotation information integrated with abundance of the transcripts per million (TPM) showed values in the order CIP > MC > CIB (Appendix A) [53]. The species were annotated by MMseqs2 according to the Genome Taxonomy Database (GTDB, https://gtdb.ecogenomic.org/), National Center for Biotechnology Information (NCBI, www.ncbi.nlm.nih.gov), Methane cycling database (McycDB, www.github.com/qichao1984/MCycDB) and Nitrogen cycling database (NCycDB, https://github.com/qichao1984/NCyc), etc.

The alpha diversity indexes (Chao1, Goods coverage and Simpson) of the species composition in the three groups significantly differed (*p* < 0.05), indicating that the CIP and CIB had higher microbial diversity than under MC (Appendix A). The PCoA results from the beta diversity analysis displayed strong differences in microbial diversity among the three groups (Appendix A), and the microbial diversity composition of CIP and CIB was more similar at the phylum level (Appendix A). For the three groups, the microbial composition was 96.2% bacteria, and 3.8% eukaryotes, archaea and viruses. In this study, the main differences at the phylum level among the three groups were concentrated in Proteobacteria, Acidobacteriota, Actinobacteriota, Myxococcota, Gemmatimonadota, Chloroflexota, Methylomirabilota, Dormibacterota, Nitrospirota, and Desulfobacterota. Proteobacteria (43.68–48.68%), Acidobacteriota (18.88–21.55%), Actinobacteriota (11.22–23.31%), Myxococcota (2.39–3.88%), Gemmatimonadota (1.55–2.58%), and Chloroflexota (1.55–2.58%) were the dominant phyla (Figure 3b). LEfSe analysis showed that the LDA threshold was 3.74, and both LEfSe analysis and random forest analysis showed that Sulfotelmatobacter, Acidoferrales, Acidoferrum, Pseudolabrys, Methyloceanibacter, Myxococcota, and Rhziobiales,, etc., were significant in the composition of the microbial community (Figure 3c). Under CIP, CIB, and MC at the genus level, there were 6,420, 6,181, and 6,378 different species (Figure 3a), including those belonging to Usltatibacter, Pseudolabrys, Palsa, Acidoferrum, and Sulfotelmatobacter (Figure 3d).

The annotation results showed that the main microbial phyla in the rhizosphere soil of *C. fungigraminus* with methane and nitrogen metabolic abilities were Acidobacteriota (19.20%), Actinobacteriota (14.31%), Chloroflexota (5.40%), Planctomycetota (3.93%), Myxococcota (4.44%), Desulfobacterota (1.82%), Verrucomicrobiota (3.66%), Desulfobacterota (1.85%), Methylomirabilota (1.57%), and Gemmatimonadota (1.83%). Under MC, the unique phylum was Nitrospirota (2.2%), and under CIP and CIB, the unique phyla were Bacteroidota (2.2%) and Dormibacterota (1.4%) (Figure 4a,b). According to the Kruskal–Wallis test, the abundances of Acidobacteria, Sphingomicrobium, Usitatibacter, Sulfotelmatobacter, and Betaproteobacteria were higher under CIB (*p* < 0.05), and the abundances of Acidoferrum and Bradyrhizobium were higher under CIP (*p* < 0.05). The abundances of Methylomirabilota and Pseudolabrys were significantly (*p* < 0.05) higher under MC than in the other treatments (Figure 3d).

### 3.4. Methane Cycle and Nitrogen Cycle Functional Gene Analysis

For all the samples, 189 gene family enzymes were analyzed in the MCycDB (https://github.com/qichao1984/MCycDB) and were found to be mainly associated with 10 pathways, including the serine cycle, aceticlastic methanogenesis, the anaerobic oxidation of methane, the oxidation of methane and C1 compounds, the ribulose monophosphate (RuMP) cycle, and the central methanogenic pathway. The most abundant genes included *acd*A (the average TPM was 5722, the same as below), *acs* (3075), *gly*A (2590), *fol*A (2254), *pfp* (1580), *frm*A (1548), *mcl* (1386), *met*F (925), and *hdr*D (859) (Figure 4c and Appendix A). The main enriched pathways were the serine cycle, aceticlastic methanogenesis, the oxidation of formaldehyde, the RuMP cycle, the oxidation of formate, and central methanogenesis. Fifty-nine gene families encoding enzymes were analyzed in the NCycDB (https://github.com/qichao1984/NCyc) and were associated with assimilatory nitrate reduction, denitrification, dissimilatory nitrate reduction, nitrification, nitrogen fixation and organic degradation and synthesis. Of these, *gln*A (2808), *gls*A (756), *nas*A (673), *nrf*C (684), *nos*Z (463), *nir*K (390), *hao* (86) and *nif*H (29), which are enriched in organic degradation and synthesis, denitrification, dissimilatory nitrate reduction, assimilatory nitrate reduction, nitrification, and nitrogen fixation, were the most abundant (Figure 4d and Appendix A).

As shown in Figure 5, the methane cycle of soil microbes under all treatments mainly included the serine cycle and RuMP cycle, and the differentially expressed genes involved in this cycle included *gly*A (2575), *mdh* (1335), *ser*A (1196), *eno* (991), *mtk*A (411), *ppc* (294) and *apg*M (34). All genes except *ser*A (1196), *fba*B (78) and *apg*M (34) had a higher expression level in CIB and were significantly different from those under MC (*p* < 0.05) (Figure 5a). In the *C. fungigraminus* rhizosphere soil microbial nitrogen cycle pathway, under CIP and CIB treatments, NR (455), *nir*A (415), and *ure*C (318) participated in nitrogen metabolism to reduce NO_3_^−^ and NH_4_^+^, by 1.10 to 1.26 times more than under MC treatment. The genes *nor*B (107) and *nos*Z (465) encode functions participating in the reduction in NO to N_2_O; of these, the *nor*B gene abundance in CIB was 1.21 times higher than that under MC (*p* < 0.05). The abundances of the *nar*G (334) and *nir*K (384) genes were highest under MC during the process of NO_3_^−^ reduction to NO, 1.26 and 1.02 times higher than those under CIB (*p* < 0.05), and the main pathways were denitrification, assimilatory nitrate reduction and organic degradation and synthesis. During nitrogen fixation, the *nif*H (29) abundance was 1.46 times higher under CIP than under CIB but did not differ from that under MC (*p* > 0.05, Figure 5b).

### 3.5. Associations among Soil Environmental–Microbial Traits and C. fungigraminus Yield

According to redundancy analysis (RDA), environmental factors had a significant impact on soil microbial diversity (the permutation test *p* value was 0.002), with higher similarity in microbial community composition observed between CIP and CIB. SMBC, SOM, AMN, pH, S-POD and soil amino sugar had positive effects on CIP and CIB, while S-CAT, S-NiR and WC had negative effects (Figure 6a). According to the net heatmap, the environmental factors in *C. fungigraminus* seedling mono-planting and the corresponding key genes in the methane cycle or the nitrogen cycle are without any association (Mantel test, *p* > 0.05; Figure 6b); However, pH, SOM, NIN, S-CAT, S-NITS, SMBC, and AMN in the *C. fungigraminus* rhizosphere soil interplanting model have significant effects on the methane cycle and nitrogen cycle key genes (Mantel test, *p* < 0.05; Figure 6c). The PLS-PM results showed that the MC, CIP, and CIB goodness-of-fit (GOF) values were 0.556, 0.631 and 0.574, respectively. Under CIP and CIB, soil physical and chemical properties had positive effects on soil microbial abundance and amino sugar formation. Under CIP, soil microbial diversity was able to promote the expression of the soil methane cycle and nitrogen cycle genes, thus promoting the serine cycle and nitrification (Figure 6d–f).

## 4. Discussion

Interplanting *C. fungigraminus* with potatoes or broad beans had previously been studied, because the interplanting of two crops has widely been accepted as a sustainable agricultural strategy worldwide, and it plays an important role in improving soil fertility, regulating the rhizosphere microbial community, inhibiting or directly killing pathogens and reducing disease occurrence [54,55]. Our experiments have shown that SMBC, SOM, AMN, pH and soil amino sugars had a positive effect on the rhizosphere soil of CIP and CIB as well as enhanced soil nitrogenase, nitrite reductase, and peroxidase activities. Moreover, CIP improved the root vitality and crude protein. These two interplanting models also improved the microbial composition and diversity in the rhizosphere soil of *C. fungigraminus* seedlings.

### 4.1. Effect of Interplanting C. fungigraminus with Crops on Soil Physicochemical Properties

Soil organic matter, total nitrogen levels and amino sugars are important indices for evaluating soil fertility and quality. Although soil organic matter and total nitrogen account for only a small fraction of the total soil volume, they play a vital role in balancing soil fertility, environmental protection, and sustainable agricultural production [56]. Interplanting also has a protective effect on carbon and nitrogen in the soil, and was shown to improve soil fertility [57,58]. The increased organic matter under different treatments in this study indicated that it played a vital role in soil fertility under the strip relay interplanting system. Soil moisture and organic matter are the main environmental factors affecting the amino sugar content, and high soil nitrogen content is conducive to the formation of amino sugars. Soil amino sugars are biomarkers that characterize microbial residues, accounting for 5–12% of the soil organic nitrogen and 2% of the organic carbon components [39,59,60]. In this study, the levels of NIN, SMBC, and SOM in all planting modes were improved compared to those in OS. Compared to those under MC, the levels of AMN, SMBC and SOM under CIP increased, as did the levels of AMN, NIN, SMBC and SOM under CIB. Furthermore, the content of CP and CF in *C. fungigraminus* seedlings provide better silage under CIP. The content of soil amino sugars varies in different habitats due to the microbial community structure, abundance, and metabolic capacity. The total amount of amino sugars has a negative response to S-NAG enzyme activity under lower nitrogen treatment [41]. There are generally more amino sugars present in the different plant species and surface litter layer [61]. Compared to the original soil, both monoculture and interplanting modes indicated that different plant diversity performance enhanced total amino sugars in the soil.

Nitrogen-fixing crops (NFCs), such as soybean, broad bean and peanut, are often used as intercrops in interplanting patterns. NFCs are able to fix N from the atmosphere, improve soil fertility, provide nutrients for plant growth, and change the levels of other nutrients in the soil; the variation trend of nitrogen content in soil and leaves was consistent under interplanting and fertilizer reduction [62]. The interplanting of Gramineae plants and legumes improves the absorption and transformation of nutrients and increases crop yield [63,64]. Nitrogen is an important element for nutrient synthesis. Our results showed that the change in nitrogen content was consistent with the trend in the three interplanting modes and consistent with the corresponding *C. fungigraminus* yield. Therefore, it can be inferred that the interplanting method affects the uptake of nutrients in *C. fungigraminus* seedling roots, resulting in yield differences, which is consistent with the results of previously published studies [57,62,65].

### 4.2. Effect of Interplanting C. fungigraminus/Crops on Microbial Diversity and Structure

Since microbes are the key components and regulators of the rhizosphere microbial community, it is necessary to investigate the difference in the microbial community of the *C. fungigraminus* rhizosphere between interplanting and monoculture and, in particular, to explore the beneficial bacteria associated with yield. Plants can provide large amounts of energy substances to soil through rhizospheric microorganisms and create a unique microenvironment for bacterial colonization in the rhizosphere soil around the roots [66]. In addition, plants can recruit microbes from other plants around them [67]. A previous study showed that sugarcane and soybean interplanting changes the soil fungal community structure and increases the abundance of Trichoderma, which help to protect against phytopathogens, and decreases the potential phytopathogens Fusarium and Curvularia, compared to monoculture [11,65]. Mulberry and alfalfa interplanting increases microbial metabolic activity, the richness index, and the dominance index, and changes the carbon source types for soil microbial utilization [68]. Based on our results, the microbial composition included 99.13% bacteria under the three treatments; of these, Proteobacteria, Acidobacteriota, Actinobacteriota, Myxococcota, Gemmatimonadota, Chloroflexota, Methylomirabilota, Dormibacterota, Nitrospirota, and Desulfobacterota were the dominant phyla. Among CIP, CIB, and MC, the different genera included Usltatibacter, Pseudolabrys, Palsa, Acidoferrum, and Sulfotelmatobacter. Under MC, the unique phylum was Nitrospirota, and under CIP and CIB, the unique phyla were Bacteroidota and Dormibacterota. According to the Kruskal–Wallis test, the abundances of Acidobacteria, Sphingomicrobium, Usitatibacter, Sulfotelmatobacter, and Betaproteobacteria were highest under CIB, and the abundances of Acidoferrum and Bradyrhizobium were higher under CIP. Under MC, the abundances of Hyphomicrobiales and Pseudolabrys were higher than those in the other groups. It has been reported that, in the rhizosphere, Actinobacteria, Proteobacteria, and Firmicutes are the key bacterial phyla that affect crop health and contribute to disease suppression [69,70]. Decreases in the abundances of Firmicutes and Actinobacteria in the tomato rhizosphere aggravate bacterial wilt disease [71]. In addition, disease-resistant varieties often tend to recruit more Actinobacteria, Proteobacteria, and Firmicutes in their rhizosphere [72]. Soil microorganisms can be considered bioindicators of the system. The soil presents conditions for the development and population growth of microbes, which means that the soil is well-structured chemically and physically. The simplification of ecosystems and soil degradation may decrease the density and diversity of the soil biota [73,74]. Taken together, our results provide further evidence that interplanting can change the structure of the soil microbial community and increase the amounts of bacteria that assist disease resistance in interplanting soil. We also inferred that *C. fungigraminus* interplanting with potatoes or broad beans may enhance crop disease resistance through soil microbial community changes.

### 4.3. Effect of Interplanting C. fungigraminus/Crops on Methane and Nitrogen Cycling

Methane is an important energy source and a greenhouse gas that is less abundant than CO_2_ on Earth. From 1750 to 2000, the concentration of methane in the atmosphere increased by approximately 150%, causing 20% of the total global warming effects due to greenhouse gasses. Recently, the concentration of methane has continued to increase, increasing by 4.2% from 2006 to 2017 [75], and has reached 2.5 μL·L^−1^. The emission of methane from farmland soil into the air has become one of the main processes of carbon flow in agricultural ecosystems. In our study, Methylomicrobium, a genus of a salt-tolerant methane-oxidizing bacterium [76], was found under all treatments, and the relative abundances under MC, CIP and CIB were 2.20%, 1.25%, and 1.05%, respectively. The relative abundances of Gammaproteobacteria, which are type I methane-oxidizing bacteria that assimilate formaldehyde mainly through the RuMP pathway, under MC, CIP and CIB, were 9.54%, 12.07%, and 12.93%, respectively. The relative abundances of Alphaproteobacteria, which are type II methane-oxidizing bacteria that assimilate formaldehyde through the serine pathway [77], under MC, CIP and CIB, were 20.42%, 22.71 and 21.96%, respectively. Under CIP and CIB, the methane cycle was mainly mediated by the serine pathway (Figure 7a), and the abundances of *frm*A, *eno*, *ppc*, *mdh* and *mtk*A in both models were significantly increased compared to those under MC. Therefore, *C. fungigraminus* interplanting with potatoes or broad beans was able to effectively promote the absorption of methane. In fact, scientists have controversial views regarding the effect of nitrogen on methane sequestration. On the one hand, in the early 1980s, Steudler et al. [78] reported that high concentrations of soil ammonia nitrogen in a temperate forest have an inhibitory effect on methane sequestration. It has been confirmed that the main reason for this is oxidation competition between CH_4_ and NH_4_^+^ [79,80]. According to Shrestha et al. [81], adding ammonium sulfate inhibits methane absorption in forests, pastures, irrigated rice fields, and wheat millet rotation fields in Nepal. These similar results were confirmed in ecosystems such as forests, farmland, and grasslands in the 1990s [82,83,84]. On the other hand, in recent years, there have been some completely opposite results. Both Zhao et al. [85] and Yang et al. [86] demonstrated that nitrogen had no effect on methane absorption in the Qinghai Tibet Plateau, as well as in a temperate deciduous forest in North China. According to our research, it is shown that the short-term interplanting process of *C. fungigraminus* with potatoes or broad beans in the field increases the soil ammonia nitrogen content to a certain extent, but it is much lower than the ammonia nitrogen content in forests and wetlands, which can also promote methane sequestration.

Plant rhizosphere soil is an important site for energy and material exchange, especially for meeting plant nitrogen requirements for growth and development. The nitrogen cycle is the most basic cycle in soil ecosystems, including biological nitrogen fixation, mineralization, nitrification and denitrification. MC, CIB and CIP had similar nitrogenase activities, while the *nif*H abundance was higher under MC than under the other treatments and had a positive correlation with soil amino sugars. In the global terrestrial ecosystem, 95% of nitrogen transformation is carried out in the plant–microbe–soil system. Plants and microbes absorb and decompose nitrogen compounds such as nitrate and ammonium from the soil and combine nitrogen-containing compounds within them to produce amino acids so that the nitrogen can enter the organism and be absorbed and utilized. Previous research has shown that nitrogen-fixing bacteria in the roots, stems and rhizosphere soil of *C. fungigraminus* have high nitrogen-fixing enzyme activities [87,88]. Nitrification, as the central link of the soil nitrogen cycle, connects nitrogen fixation and denitrification, which not only determines the availability of nitrogen in soil but is also closely related to water pollution caused by nitrate leaching [89]. SMBC, SOM, AMN, pH, and S-POD have a positive influence on the microbial community and on key genes in the methane cycle and the nitrogen cycle (Figure 6). As shown in Figure 6c and Figure 7b, the interplanting mode of *C. fungigraminus* promoted the abundance of *nir*A and *ure*C genes involved in the nitrification reaction, increased the content of Glu and Gln in the soil, and promoted the seedling growth, root activity and yield of *C. fungigraminus*. 

### 4.4. Management Advantages of Interplanting C. fungigraminus/Crops 

The mode of *C. fungigraminus* interplanting with potatoes or broad beans crops can alter the physicochemical properties, microbial composition, and diversity of rhizosphere soil, as well as affect the methane and nitrogen metabolism of the soil microbial environment. Under a short-term influence, the monoculture of *C. fungigraminus* is able to increase soil moisture, organic matter, and enzyme activities, and does not reduce soil fertility, even though planting *C. fungigraminus* does not comply with land use policies in southern China. Interplanting *C. fungigraminus* with broad beans and potatoes has been shown to improve soil AMN, NIN, SOM, and SMBC, as well as promote methane sequestration, and improve the farmland economic benefits and feed quality. Based on food security, this model was able to vigorously develop grassland farming and animal husbandry, reduce the area of fallow fields in winter, and may become an important way to help rural revitalization and increase the income of farmers in China.

## 5. Conclusions

In southeast China, *C. fungigraminus* can be planted as high-quality perennial forage grass. Here, we demonstrated a new model of interplanting *C. fungigraminus* with potatoes or broad beans, which can improve the fertility of rhizosphere soil and root activity and crude protein of *C. fungigraminus* seedlings. Meanwhile, interplanting *C. fungigraminus* with potatoes and broad beans changes the composition and abundance of microbes in the rhizosphere soil of *C. fungigraminus* and promotes the methane cycle, nitrification, and the formation of soil amino sugars. Moreover, interplanting *C. fungigraminus* with potatoes or broad beans did not significantly reduce the yield of *C. fungigraminus* but increased the total income per unit land area and improved the utilization of idle fields in winter. Furthermore, the future interplanting models of *C. fungigraminus* can include other crop types, in order to improve land use efficiency, sustainable agricultural development and rural revitalization.

## Figures and Tables

**Figure 1 microorganisms-12-01665-f001:**
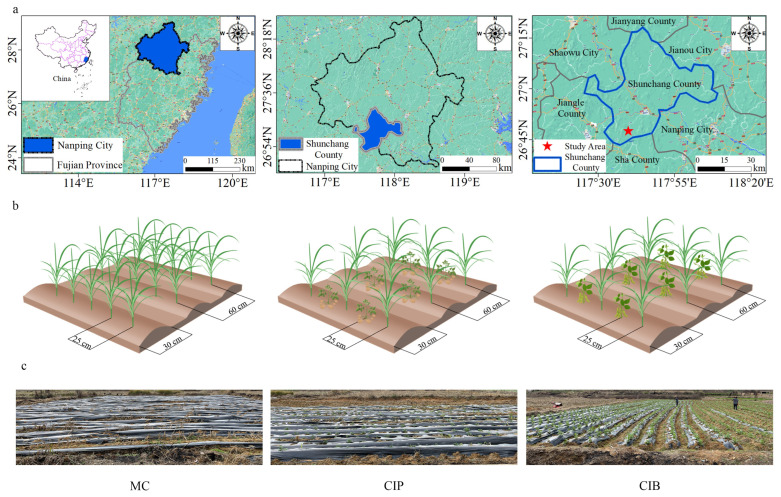
Study area (**a**) and experimental treatments (**b**,**c**). Note: *C. fungigraminus* monoplanting (MC), *C. fungigraminus*–potato interplanting (CIP) and *C. fungigraminus*–broad bean interplanting (CIB).

**Figure 2 microorganisms-12-01665-f002:**
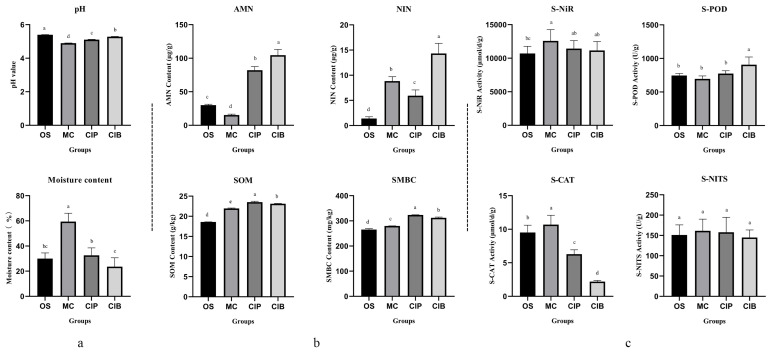
Effects of different planting modes on the physical properties including pH and moisture content (**a**), chemical properties including ammonium nitrogen (AMN), soil nitrate nitrogen (NIN), soil organic matter (SOM), and soil microbial biomass carbon (SMBC) (**b**), and enzyme activities including soil catalase (S-CAT), soil peroxidase (S-POD), soil nitrogenase (S-NITS), and soil nitrite reductase (S-NiR) (**c**) of the rhizosphere soil of *C. fungigraminus* seedlings. Significant differences at 95% (*p* < 0.05) are indicated by different lowercase letters.

**Figure 3 microorganisms-12-01665-f003:**
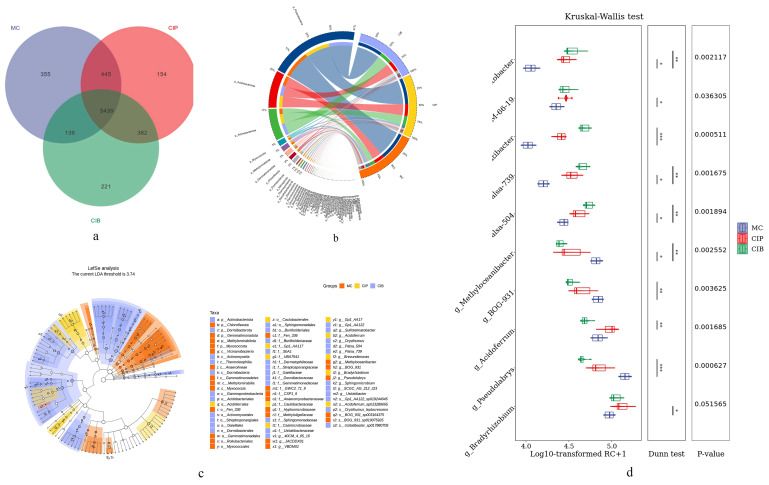
(**a**) Venn diagram of microbial community composition at the genus level; (**b**) Circos map of microbial community composition at the phylum level in three groups (top 50); (**c**) LEfSe analysis of microbial community composition at the phylum level in three groups (top 50); (**d**) difference in microbial community abundance at the genus level (top 10). Gene abundance is represented by the number of reads mapped on the gene, which is called reads count (RC). Significant differences at 95% (*p* < 0.05), 99% (*p* < 0.01) and 99.9% (*p* < 0.001) are indicated by “*”, “**”, and “***”.

**Figure 4 microorganisms-12-01665-f004:**
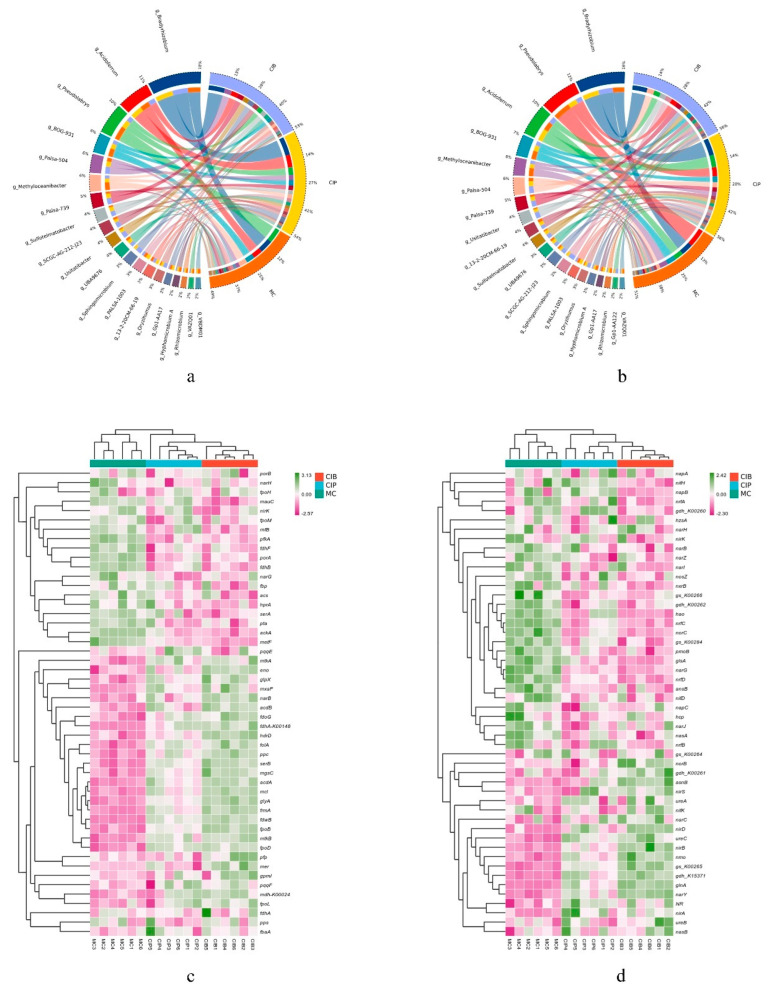
Circos diagram of average microbial community composition at the genus level for the methane cycle (**a**) and nitrogen cycle (**b**); heatmap of the top 50 functional genes’ abundance involved in the methane cycle (**c**) and nitrogen cycle (**d**).

**Figure 5 microorganisms-12-01665-f005:**
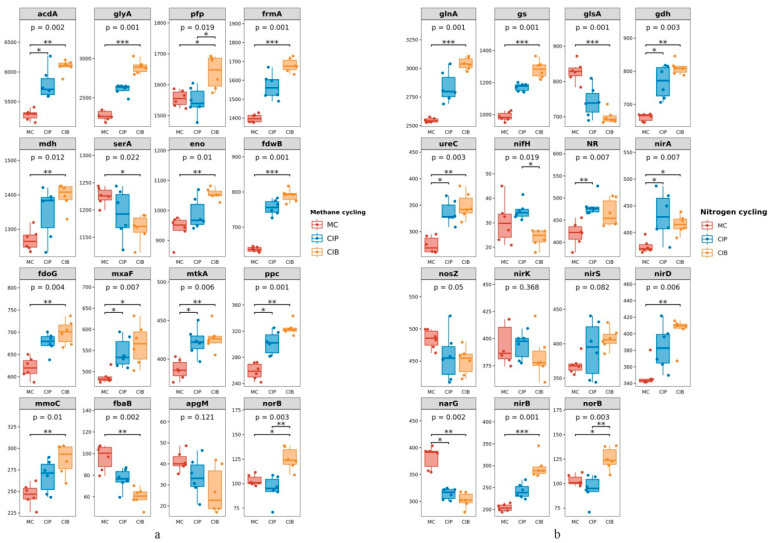
Abundance of methane cycle (**a**) and nitrogen cycle (**b**) genes in different processes. “*”, “**”, and “***” indicate *p* < 0.05, *p* < 0.01, and *p* < 0.001, respectively. (**a**) *acdA* is involved in aceticlastic methanogenesis; *gly*A, *mdh*, *ser*A, *eno* and *mtk*A, *ppc* and *apg*M are involved in the serine cycle; *pfp* and *fba*B are involved in the RuMP cycle; *frm*A, *fdw*B, *fdo*G, *mxa*F, and *mmo*C are involved in the oxidation of formate and the oxidation of methane and C1 compounds; (**b**) *gln*A, *gs*, *gls*A, *gdh* and *ure*C are involved in organic degradation and synthesis; *NR* and *nir*A are involved in assimilatory nitrate reduction; *nos*Z, *nir*K, *nir*S, *nir*D, *nar*G, *nor*B and *nir*B are involved in dissimilatory nitrate reduction; *nif*H is involved in nitrogen fixation.

**Figure 6 microorganisms-12-01665-f006:**
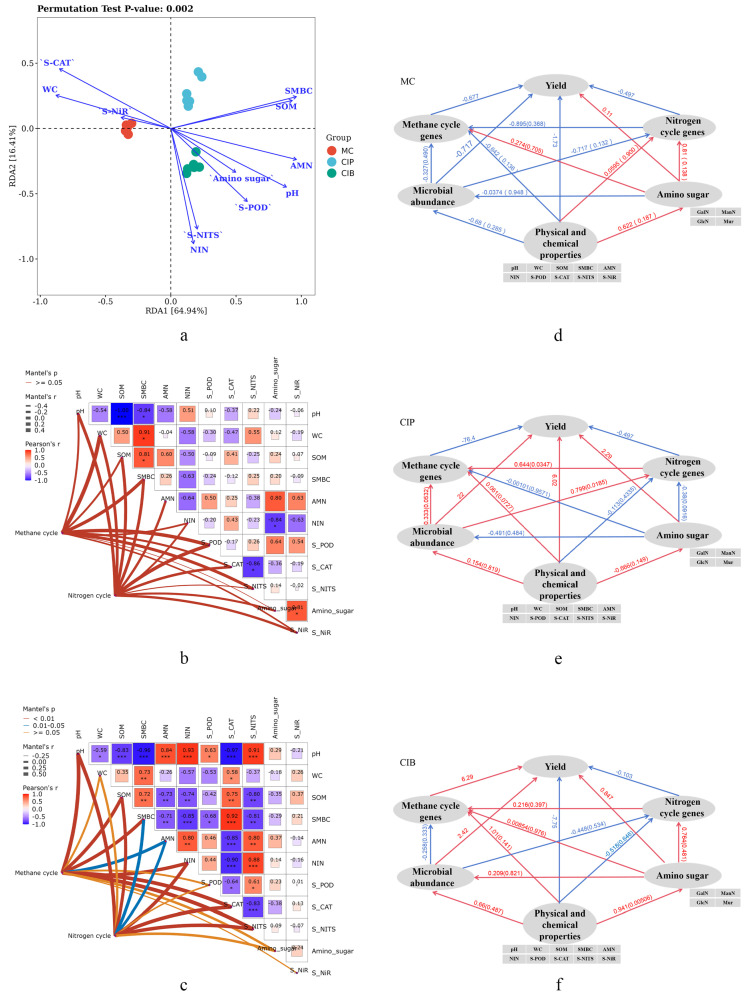
(**a**) RDA of microbial abundance and soil environmental factors in *C. fungigraminus* seedlings; (**b**) the net heatmap for methane cycle and nitrogen cycle key genes and soil environmental factors in MC; the horizontal and vertical axes represent various environmental factors: red blocks within the grid indicate a positive correlation between environmental factors, while blue blocks indicate a negative correlation. The size of the colored block represents the absolute value of the correlation coefficient. The line thickness represents the correlation between methane cycle and nitrogen cycle genes and environmental factors, and the line color represents Mantel’s significance; (**c**) the net heatmap for the methane cycle and nitrogen cycle key genes and soil environmental factors in CIP and CIB; (**d**–**f**) Direct associations among soil physical and chemical properties, microbial abundance, functional genes and *C. fungigraminus* yield in 3 planting treatments. Significant differences at 95% (*p* < 0.05), 99% (*p* < 0.01) and 99.9% (*p* < 0.001) are indicated by “*”, “**”, and “***”.

**Figure 7 microorganisms-12-01665-f007:**
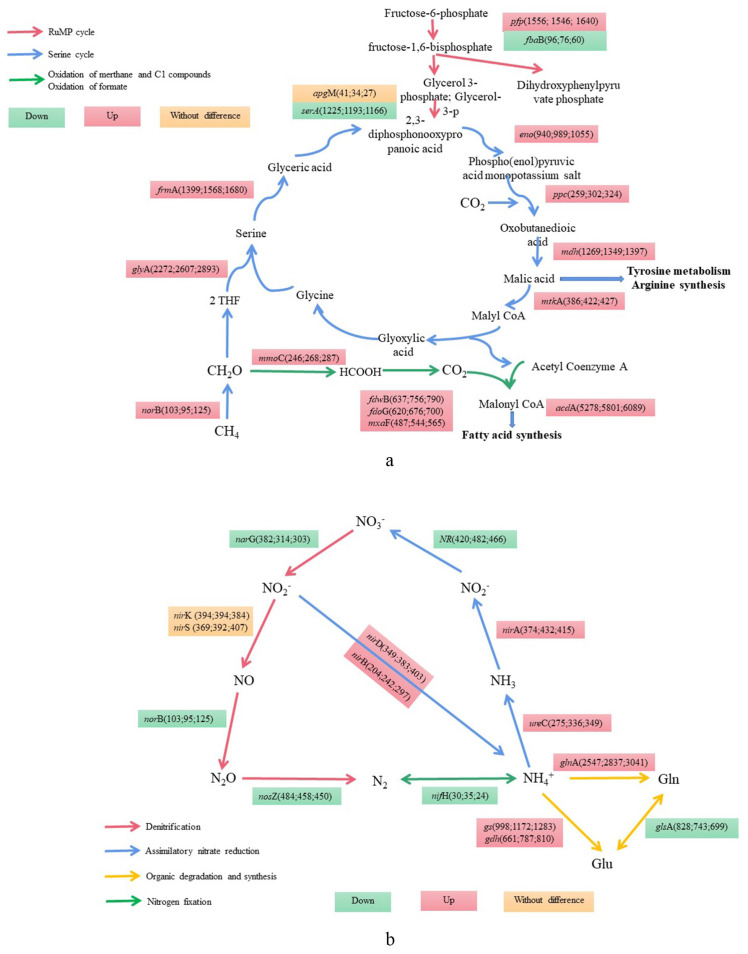
Functional genes involved in the methane cycle (**a**) and the nitrogen cycle (**b**) in the rhizosphere soil microbial of *C. fungigraminus* interplanted with crops.

**Table 1 microorganisms-12-01665-t001:** Effect of *C. fungigraminus* growth on different planting modes (means ± standard deviation).

Treatments	Total Yield (Ton·ha^−1^, Fresh Weight, in July 2023)		Total Value (CNY)	*C. fungigraminus* Seedling (in April 2023)							
	*C. fungigraminus*	Crops		TS (mg·g^−1^)	CP (g·kg^−1^)	CF (mg·g^−1^)	Height (cm)	Leaf Length (cm)	Tiller Number	Dry Root Weight (g)	Root Activity (mg·d^−1^·h^−1^)
MC	189.72	-	20,355	575.08 ± 7.95 a	83.30 ± 3.23 a	290.74 ± 4.54 c	97.53 ± 15.20 a	77.68 ± 10.48 a	2.67 ± 1.21 a	15.33 ± 3.48 a	131.56 ± 47.46 b
CIP	122.10	2.92	23,412	465.39 ± 6.15 c	92.82 ± 0.56 b	309.56 ± 6.59 b	85.17 ± 17.61 a	69.75 ± 14.20 a	2.83 ± 0.98 a	5.69 ± 3.68 b	274.43 ± 108.66 a
CIB	135.70	0.50	20,600	519.92 ± 6.93 b	73.49 ± 1.32 a	329.36 ± 4.35 a	92.97 ± 15.99 a	72.73 ± 13.83 a	3.83 ± 1.83 a	5.88 ± 2.62 b	66.98 ± 10.06 b

Note: According to the market and https://www.cnhnb.com/ (2023), potato is priced at CNY 2000·ton^−1^, broad bean at CNY 8000·ton^−1^, and fresh *C. fungigraminus* at CNY 120·ton^−1^. Total sugar (TS), crude protein (CP) and crude fiber (CF). Significant differences at 95% (*p* < 0.05) are indicated by different lowercase letters.

## Data Availability

The original contributions presented in the study are included in the article/Appendix A, further inquiries can be directed to the corresponding author.

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
