# Peer review of "Short-Term Effects of Cenchrus fungigraminus/Potato or Broad Bean Interplanting on Rhizosphere Soil Fertility, Microbial Diversity, and Greenhouse Gas Sequestration in Southeast China"

_microorganisms, 2024, doi:10.3390/microorganisms12081665_

Round 1

Reviewer 1 Report

Comments and Suggestions for Authors

Concerning the abstract, I would also include at least some description of your rationale for the project, why study this particular grass, your experimental setup and environment, and some idea of the methods. 

In lines 57-61, you indicate that Chinese agricultural systems have used intercropping for a long time, with many advantages. Although several farm systems worldwide have used this kind of system, I highly doubt ancient farmers considered these advantages you mention since research has only recently (as seen in your references) discovered most of them. Please take care of this kind of mistake. 

The experimental setup is at least very difficult to understand. For instance, what was the experimental design and number of reps? You indicate seeding both broad beans and potatoes in December, which would be the winter at this latitude. Is that the case for a locale with 60 days of frost (since you tell us it has 305 frost-free days)? How did you measure soil nitrogenase?

Your description of the system does not allow a proper evaluation of the statistical analysis, but why did you say that you anova OR mantel test were performed? Your results indicate you compared post crop and pre crop data, without any explanation of how you did that. 
Table 1 includes variables that were not described in the methods or that at least I could not identify there, such as total value. You did not include any statistical analysis information for the first three variables. 

These problems are somewhat repeated for the remaining results. 

The first paragraph of the discussion is more of an introduction than a discussion since your results are not mentioned. 

The remainder of the discussion is very hard to follow, with very long paragraphs discussing several different topics with little connecting information, and much of the literature being cited as a comparison but not offering possible explanation of your results. 

Comments on the Quality of English Language

I recommend a very careful language review. For example, on the abstract you affirm that: "Compared to MC, soil microbial biomass carbon (SMBC), soil organic matter (SOM), ammoniacal nitrogen (AMN), pH and soil amino sugars had a positive effect on rhizosphere soil of CIP and CIB as well as enhanced soil nitrogenase, nitrite reductase, peroxidase activities (P < 0.05)." which doesn´t make sense, at least to me. I believe you wanted to mention that these variables were increased under the consortiums rather than solo cropping, but what is said in the text is not that, at least in my reading. I also recommend you break your paragraphs into a single central idea each and reduce their length to increase reading comprehension. For example, the first paragraph of the introduction spans at least four major topics over close to 40 lines. 

Reviewer 2 Report

Comments and Suggestions for Authors

Manuscript entitled. "Short-Term Effects of Cenchrus Fungigraminus/Potato or Broad Bean Interplanting on Rhizosphere Soil Fertility, Microbial Diversity, and Greenhouse Gas Sequestration in Southeast China" is interesting and may be of interest to specialists in the discipline represented. However, in order for it to be publishable, I suggest that the following comments be considered.

1. Introduction, Materials and Methods, Discussion and References - are not cited in accordance with the requirements of the journal.

2. Materials and methods - this chapter should be more clearly described. Citing national methodological literature is not sufficient. The study procedure should also be described more precisely. The factors studied should be listed more precisely so that the abstract and conclusions can be clearly addressed. Please indicate how many plots were used in the study. Line 189 - was the column really 30m long????

3 Results - Figure 2 - some results are given in g soil and others in kg soil. Perhaps it would be better to give them all in terms of 1 kg soil. The quality of Figures 3 and 4 should be improved. Consideration should also be given to improving the quality of other figures.

4 Discussion - The chapter is well written except for the misquoting of literature.

Reviewer 3 Report

Comments and Suggestions for Authors

I read the manuscript I found some areas in which I would have appreciated greater clarity ie. there are minor technical errors:.

The abstract gives a summary of the results.

Line 32- Latin name plant should be writing in italic form- Cenchrus fungigraminus, and so on throughout the manuscript

Introduction part provides o good and comprehensive overview of the topic and background.

Lines 68 and 69- It is well known that soil 68 microbial diversity has a critical role in maintaining the sustainability of agricultural pro- 69 duction systems (Sammauria et al., 2020)- put this sentence later

Line 70- Plants harbour numerous microbial communities in the rhizosphere ecosystem.

- this is not well written, - the plant does not harbour, but the rhizosphere microbial community, this sentence should be corrected

Lines 86-88

Therefore, the recruitment of ni- 86 trogen-fixing bacteria in symbiotic or nonsymbiotic relationships helps host crops obtain  nitrogen directly from the atmosphere to meet their nutritional needs (Matos et al., 2021;- this sentence should be corrected, in a symbiotic community there is a macro and a micro symbiont, there is no host plant...

In the section Introduction the reference Mussatto, 2014 has been used in two sentences that can be corrected and combined into one..

The lat paragraph in Introduction should be written more clearly, highlighting the goal of the research, please

The methods are appropriate, and they been correctly described and applied.

The Results and discussion parts is well structured and contains all results, which are clearly described in figures and tables. The authors are describe and analyzes their results and linked them with already published literature.

In Conclusions the authors showed new model of interplanting C. fungigraminus with

potatoes or broad beans, which can improve the fertility of rhizosphere soilThis plant species can include other crop types, in order to improve land use efficiency, sustainable agricul-tural development and rural revitalization.

It should not write in the first person, please
